# Early development of primary care networks in the NHS in England: a qualitative mixed-methods evaluation

Sarah Parkinson [1], Judith Smith [2], Manbinder Sidhu [2]

## ABSTRACT

**Objectives** Primary care networks (PCNs) were introduced in the National Health Service (NHS) in England in 2019 to improve integrated care for patients and help address financial and workforce sustainability issues in general practice. The purpose of this study was to collect early evidence on their implementation and development, including motivations to participate and what enables or inhibits progress. This paper considers the core characteristics of PCNs, and how this informs their management.

**Design** A qualitative mixed-methods rapid evaluation was conducted across four case study sites in England, informed by a literature review and stakeholder workshop. Data collection comprised interviews, non-participant observation of meetings, an online survey and documentary review.

**Results** General practitioners (GPs) are motivated to participate in PCNs for their potential to improve patient care, enable better coordinated services and enhance financial and workforce sustainability within primary care. However, PCNs also have an almost mandatory feel, based on the national policy context and significant financial incentives associated with joining them. PCNs offer potential to bring GPs together to work towards common goals, deliver national priorities and respond rapidly to local needs.

**Conclusions** PCNs face similar challenges to other meso-level primary care organisations internationally, as they respond to local and national priorities and operate in a context of multiple goals and interests. In managing these organisations, it is important to find a balance between local and national autonomy, decision making and control.

## Strengths and limitations of this study

► As a rapid evaluation, this study responds to current policy-relevant questions about the early development of primary care networks (PCNs), and developments in how primary care is delivered in the National Health Service (NHS) in England.

► The qualitative approach provides insights into why general practices participate in PCNs and the experience of implementation.

► The mixed-methods approach to this evaluation allows data to be triangulated between sources and ensures that a broad range of perspectives is captured.

► The use of a theoretical framework to interpret the findings from this evaluation helps contextualise them within the wider literature, and understand what this evaluation means for other meso-level primary care organisations internationally.

► This evaluation provides an insight into the early development and implementation of PCNs, along with information about their initial response to the COVID-19 pandemic. Although the study reflects on how PCNs will continue to develop, for example, in response to new policies in the English NHS, definitive conclusions about the impact of PCNs were outside the scope of this study.

[1]Health and Wellbeing, RAND Europe, Cambridge, UK
[2]Health Services Management Centre, University of Birmingham, Birmingham, UK

**Correspondence to**
Professor Judith Smith;
j.a.smith.20@bham.ac.uk

## INTRODUCTION

Primary care networks (PCNs) are the latest in a long line of general practice collaborations in the National Health Service (NHS) in England dating back to GP fundholding and locality commissioning in the 1990s.[1] Predecessor collaborations have encompassed a wide range of arrangements, including total purchasing projects, primary care groups and trusts, practice-based commissioning, personal medical services schemes and clinical commissioning groups (CCGs).[2–4] These collaborations have ranged from informal networks to formal multi-site practice organisations and super-partnerships where GP practices merge important functions such as managing finances and contracts.[5] These other forms of collaboration have had varied aims, including improving care at a local level and delivering new services to patients, strengthening the resilience of general practice, and supporting better management in primary care, including improved financial stability.[2 6]

PCNs were introduced in 2019 as part of the NHS Long Term Plan,[7] which claimed that these new networks would create integrated and community-based healthcare, support expanded neighbourhood teams, increase workforce sustainability and deliver on a number of national priorities such as health inequalities and early cancer

diagnosis. The NHS Long Term Plan announced that at least £4.5 billion would be invested in these networks over the following 5 years. Since this time, nearly all practices in England have joined a PCN.[8] PCNs were introduced into the English NHS at a time of particular financial and workforce sustainability challenges in primary care and general practice,[9–11] which is important in understanding their goals and policy context. Key characteristics of PCNs are set out in box 1 .

> **Box 1    Key features of primary care networks (PCNs)**
>
> ► PCNs are intended to bring together groups of neighbouring (geographically contiguous) general practices, along with other primary care providers such as community pharmacists, dentists, optometrists and voluntary sector organisations.
> ► The NHS Long Term Plan specified that PCNs should cover a patient population of 30 000–50 000 patients,[7] although many now cover much larger populations (upwards of 100 000–150 000 patients)[38]
> ► PCNs receive funding on a per-patient basis for enhanced services and additional funding to support recruitment of new shared roles such as social prescribers and clinical pharmacists. This funding is distributed through their local clinical commissioning group (CCG, the local funding agency which commissions most hospital, mental health and community services in local areas in England).
> ► Each PCN is led by a clinical director, who receives funding for their role depending on population size of the network, weighted by deprivation and burden of morbidity.[14]
> ► The Direct Enhanced Services (DES) contract[28] specifies what services PCNs must provide to gain access to funding, and includes specifications for structured medication reviews, general practice support of care homes, anticipatory care for patients in the community with complex needs, early cancer diagnosis services, cardiovascular health and health inequalities. The contract sets out what local commissioners of primary care services must offer to providers participating in PCNs, and what services PCNs must deliver to receive additional funding.

One notable way in which PCNs depart from some previous forms of collaborative working is that many prior collaborations (eg, GP super-partnerships, GP federations, GP multi-funds) evolved from the ground up, meaning that local actors within primary care had taken the initiative to work together out of necessity or shared interest. In contrast, PCNs have been encouraged through national policy with significant financial incentives,[12] giving them a compulsory, top-down feel when compared with some previous forms of collaborative working, although they share this more mandated approach with primary care groups, primary care trusts and practice-based commissioning.[13] While participation in PCNs is in theory voluntary for GP practices, in reality almost all practices have interpreted them as mandatory, considering the significant levels of new funding that are distributed through PCNs.[14]

PCNs are meso-level organisations,[6] operating between formal funders or commissioners, and local GP practices. As such, they are somewhat hybrid in nature, being both national and local, and extrinsically (eg, based on policy and incentives) and intrinsically motivated (eg, based on expected benefits and desire to collaborate) through a national policy initiative as well as shared goals and interests. As meso-level organisations, PCNs share characteristics with international experiences of primary care organisations, displaying complexity in their form, objectives and ways of working,[14] and occupying a sometimes unclear position within national and local healthcare systems.[15]

As networks of healthcare professionals, there is also much to learn about PCNs from prior work on the characteristics of professionally-led networks and healthcare network management. The existing literature explores effective ways to manage and govern networks in healthcare, depending on the structure of the network and the context within which the network is functioning.[16–20] This paper contributes to this body of literature, and applies existing theoretical work on healthcare networks management to the early experience of PCNs in the NHS in England.

This analysis draws on a rapid mixed methods evaluation of the first year of operation of PCNs[13] to explore their implementation and early progress. The findings are interpreted using theory about the nature of healthcare network structure and management, drawn from work by Goodwin et al.[21] In particular, this analysis includes an examination of the characteristics that PCNs share with 'enclave networks', with a rather flat organisational structure, formed of relatively close-knit groups of professionals and seeking to have a bottom-up and locally owned sense of purpose, as well as 'hierarchical networks,' designed to undertake specific tasks as dictated through contractual and funding mechanisms that are enacted in a top-down manner on behalf of a national health system.[21]

This analysis addresses the following questions:
► RQ1: What was the rationale for GP practices to join and participate in a PCN?
► RQ2: What enabled or inhibited the early progress made by PCNs?
► RQ3: What are the core characteristics of PCNs, given their role as meso-level organisations working between local general practice and national health funders and commissioners?
► RQ4: What does this experience reveal about how to manage and prepare meso-level primary care collaborations to fulfil local and national policy expectations?

## METHODS

The rapid evaluation study explored four case study sites across England through documentary review, interviews, a survey and non-participant observations, which are described below. The full findings from this research are described in Smith et al,[13] while this article draws on additional analysis of evaluation data to address the research questions set out above, and as detailed in table 1.

## Patient and public involvement

A half-day project design workshop was undertaken in November 2018 and involved, in addition to the research

**Table 1** Summary of methods and research questions

| Study phase | Description | Research questions (RQs) |
|---|---|---|
| Rapid evidence assessment | An overview of published evidence to distil prior learning and inform the development of propositions to be tested through comparative case studies of new PCNs | RQ1 and RQ2 |
| Stakeholder workshop | A workshop led by members of the study team for relevant stakeholders (eg, academic and policy experts in the field, patient and public involvement representatives). The aim of this workshop was to clarify evidence gaps and evaluation questions of particular relevance to emerging policy on primary care networks and thus inform next steps. | RQ1 and RQ2 |
| Comparative case studies of four primary care networks | Interviews with those involved in the conceptual design, implementation of primary care networks in their respective sites and exploration of relationship with any prior GP collaboration in the case study site; analysis of key documentation (both internal and publicly shared); non-participant observation of strategic meetings; and an online survey to collate information on challenges associated with collaborative working and measuring early impact. | RQ1, RQ2, RQ3 and RQ4 |
| Analysis of findings from case studies to develop a nuanced understanding of the development of primary care networks in the NHS in England | Share and discuss findings generated from data collection from case studies. | RQ1, RQ2, RQ3 and RQ4 |

GP, general practitioner; NHS, National Health Service; PCN, Primary Care Network; RQ, research question.

team, national primary care policy officials, a patient representative, academics with experience of researching primary care organisations and policy experts in the field (N=12). The aim of the workshop was to help identify gaps in the literature, and thereby devise relevant research questions. Participants at the workshop felt a key unexplored area was the experiences of primary care collaborations in rural, as opposed to urban areas, to better understand regionally specific challenges in primary care. Furthermore, attendees were keen for researchers to investigate sites where it had proved challenging to sustain primary care collaborations, and to examine what management and organisational development skills and capacity are needed to make a PCN work effectively.

### Sampling and recruitment of case study sites

Purposive sampling was used to select sites that had not been involved in research studies or evaluations over the previous two years, and to ensure that the sample included rural sites, as well as collaborations that had previously faced challenges in sustaining joint working. Potential CCGs to approach were identified using a combination of an online search of grey literature and those that had responded to a 2017 study by the Royal College of General Practitioners.[22] Twenty-eight CCGs were contacted as potential participants from May to August 2019, and those that responded (n=7) were sent a short survey to determine whether emerging PCNs in their area met the inclusion criteria. Three case study sites were identified using this approach, and a fourth was identified through engagement with providers known to the researchers. Three case study sites were PCNs, and

one was a super-partnership with member practices also belonging to several PCNs. A short description of the four case study sites is provided in table 2 below and a summary of the sampling approach is illustrated in figure 1.

### Data collection and recruitment

Data collection took place between September 2019 and July 2020, and was facilitated through a gatekeeper,[23] or contact point, at each case study site. A total of 29 semi-structured interviews with 25 participants were conducted using a topic guide (summarised in box 2), each lasting between 30 and 60 minutes. Participant characteristics are described in table 3. A minimum of one and a maximum of nine interviews were conducted at each of the four case study sites with both clinical and non-clinical staff, mainly with those in leadership or management positions within the PCN. Interviews were audiorecorded, transcribed verbatim using a professional transcription service and pseudonymised. Four of these interviews were follow-up interviews with PCN managers to gather information on their response to COVID-19. Interviews were completed both face-to-face and virtually (due to the onset of the COVID-19 pandemic) by JS, SP and MS and two further researchers with qualitative interviewing experience. Data saturation was achieved for themes regarding rationale for GP practices to join and participate in a PCN and what may have inhibited or enabled progress; although saturation may not have been achieved for themes focused on the trajectory of PCNs in the post-pandemic English NHS given that data collection ended in the initial phase of the pandemic.

**Table 2** Description of case study sites

| Case study site | Short description |
| --- | --- |
| Site A | PCN in a rural setting covering a patient population of 75 000 (large ageing population, mostly White British), where practices had previously worked together through an informal model of locality working. Some practices in the PCN were also involved in a super-partnership. |
| Site B | Super-partnership in a rural setting covering a patient population of 130 000 patients (large ageing population, mostly White British). Practices within super-partnership were part of four separate PCNs which also contained non-super-partnership practices. |
| Site C | PCN in an urban and semi-urban setting, covering a patient population of about 60–70 000 patients (socio-economically disadvantaged population, significant Black, Asian and minority ethnic population), where practices had previously worked together formally in a GP Neighbourhood. |
| Site D | PCN in a rural setting, covering a population of 30 000 patients (large ageing population, mostly White British), where practices previously worked together and with community teams informally. |

GP, general practice; PCN, primary care network.

Nine meetings (eg, board-level, partner-level and task group meetings) were observed across the four case study sites by SP and MS and two other researchers with experience in non-participant observations. A template was used to take notes at each meeting on the topics discussed and dynamics within each case study site, including a sociogram to visualise how meeting participants interacted with one another.[24] For both interviews and observation, participants were offered an information sheet about the study, given the opportunity to ask questions and provide informed consent prior to data collection. Lastly, gatekeepers provided access to key documents at each site, including material related to the structure of the PCN and any pre-existing GP collaboration, governance and decision-making, agendas of previous meetings and local communication activities. Information was extracted from these documents using a structured Excel template based on the aims of the evaluation.

## Synthesis and analysis

After data had been collected, the evaluation team (JS, SP and MS) participated in a half-day data analysis workshop to review data collected, discuss themes and begin systematic analysis of the data as per the framework method for data analysis described in Gale *et al*.[25] Data from interviews were analysed through deductive coding with NVivo V.12 software[26] using a codebook that had been developed by the evaluation team based on the evaluation aims, available literature on primary care collaborations and initial reading of interview transcripts. Analysis was led by SP, whereby an initial coding frame was developed based on codes arising from a sample of five transcripts by MS and SP. MS and SP coded all transcripts, and further developed codes based on subsequent transcripts and further discussions. This approach was also applied to data from non-participant observation meeting notes and documentary review template. After analysis, themes were discussed in a second half-day workshop (JS, SP and MS) with the evaluation team to synthesise evidence for each of the research questions and develop an overarching narrative summary (written by JS) of the findings.

## FINDINGS

Analysis of data from evaluation fieldwork highlights the rationale for GPs to join PCNs, what has facilitated and inhibited the early progress of these new networks, and what this means about the nature of how PCNs operate

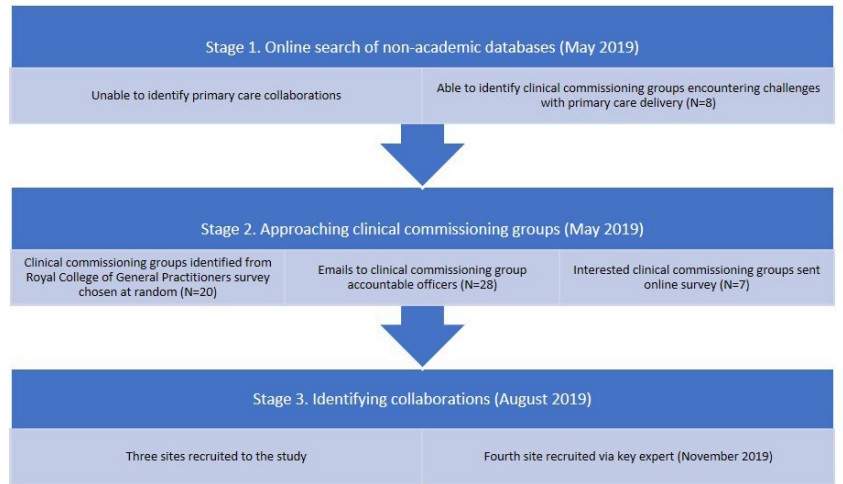
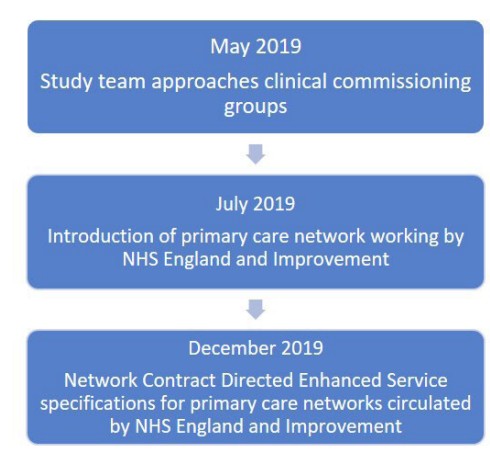

**Figure 1** Sampling approach for selection of case studies. NHS, National Health Service.

## Box 2  Interview topics

1. Models of general practice collaboration in the local area and how previous and extant collaborations relate to PCNs.
2. Specific challenges to PCN working, particularly in relation to urban and rural settings, and any practice that may have left PCNs.
3. How collaborative working in local primary care systems has evolved since introducing PCNs.
4. Nature of professional relationships within PCN.
5. Motivations to participate in PCNs.
6. Key goals and outcomes for short and medium to long term for PCNs.
7. Early impact of PCNs.

and are likely to develop longer term. The full findings of the evaluation are reported in Smith et al (2020),[13] while this secondary analysis of data from the evaluation focuses on interpreting the data in relation to healthcare network structure and management.

### Reasons for joining and participating in a PCN

There are many reasons why GP practices join and participate in PCNs, these being based on both top-down and bottom-up motivations. These reasons reflect the policy and incentive structure that led to the introduction of local PCNs within the national context of the English NHS, as well as a genuine desire to collaborate locally to ensure the sustainability of primary care and improve and enhance the services available to patients.

When asked about the reasons why their practice joined a PCN, interviewees involved in practice-level management reflected that PCNs are, in effect, perceived to be mandatory given the sizeable financial incentives associated with PCN membership. There was some sense of frustration about the perception that practices have been forced or coerced into joining PCNs, although others asserted that the national PCNs policy is based on the known efficacy of primary care in responding to incentives.

*'Most [of] my GP colleagues in other practices and within my partnership, we all were very suspicious of it and also didn't feel it was the right mechanism for delivering the resilience in*

**Table 3** Characteristics of interviewees from four case study sites

| Site | Description | No (N) |
|------|-------------|--------|
| Site A | Primary care clinical staff | 4 (Int1–4) |
| | Primary care non-clinical staff | 5 (Int5–9) |
| Site B | Primary care clinical staff | 3 (Int10–12) |
| | Primary care non-clinical staff | 3 (Int13–15) |
| | Clinical commissioning group staff | 2 (Int16–17) |
| Site C | Primary care clinical staff | 4 (Int18, 20–22) |
| | Primary care non-clinical staff | 2 (Int23–24) |
| | Clinical commissioning group staff | 1 (Int19) |
| Site D | Primary care non-clinical staff | 1 (Int25) |
| Total | | 25 |

*general practice which we need because it was being foisted on us… it was the only way we could see that we were going to get any new money coming into general practice…. I guess we thought… we might as well.'* (Int1)

Despite this focus on top-down motivations, bottom-up motivations also contributed to the desire to join a PCN. All four case study sites had a history of their GPs collaborating with one another to some extent, either through informal groupings or more formal arrangements such as super-partnerships or locality forums. Respondents involved in the management of PCNs reflected that practices typically collaborate to fill gaps in the services that single practices are able to provide, and to facilitate GPs working with community health teams, social services and the voluntary sector to provide extended care that addresses local population health needs.

*'We're only a small network, 35 000 patients in the network… I sort of see the 35 000 rather than the 3000 we've got on our list. So I'm really enthusiastic, and I want to make sure that the 35 000 are looked after, as much as my 3000'*

A clear desire to improve the sustainability of primary care was also a shared goal that motivates GP practices to work with one another within PCNs. Some interviewees also mentioned that working in collaboration across practices is attractive because of the potential for financial efficiency and sustainability by sharing back office functions, reducing duplication of administrative tasks, introducing more robust financial management processes and making it easier to recruit and retain new staff, for example, by providing more opportunities for training, education and specialisation.

The reasons for joining and participating in PCNs impact not only on individual GP practices, but also the structure of PCNs themselves. Networks can be built on the shared interests, goals and motivations of members, and also through formalised structures and top-down regulation that require or incentivise membership. In the case of PCNs, members are bonded by a blend of these structural mechanisms. National policy has prompted the forming of PCNs, but in the absence of national policy incentives, it would remain in GP practices' best interests to still collaborate with one another to provide services, improve management and realise efficiencies based on their mutual interests. This blend of motivations influences the relationship that network members have with one another, and also the place of the network within the wider health and care system.

### Local engagement and ownership of PCNs

Engagement by practices in the PCN at a local level is critical to ensuring that networks not only deliver the national priorities set for them, but also address local health needs and improve the integration of services across primary care. This is of particular importance given NHS policy direction towards new integrated care systems (ICSs).[27]

Early in the implementation of PCNs, there tended to be little engagement with the PCN below the leadership and management level among staff in constituent GP practices. At this stage, there seemed to be a sense that PCNs had not yet had much effect, and that local practices would continue to deliver services for patients and operate much as they did before PCNs.

*'Some of the staff wouldn't know that we were in a network, even though we've told them about it. If you then said about the PCN, they'd say well what's that?'* (Int6)

In some cases, this lack of engagement was reported to be exacerbated by a perception that PCNs were the latest in a long line of collaborative mechanisms set out by the NHS for GPs. Frustration was expressed about frequently changing NHS policy that disrupts extant ways of working, including activity underway to improve patient care through other forms of locally developed primary care collaborations such as federations and super-partnerships. There was also some irritation expressed by interviewees and observed in meetings about the prescriptive nature of the services required by the DES contract,[28] which further tempers local buy-in to PCNs, particularly where services specified in the DES contract are perceived as not tailored to the needs and preferences of local populations.

*'We just thought, well we've been there before. We deal with the box ticking. Get the box ticking done and then deliver what… might improve care for our patients'* (Int1)

There was also genuine enthusiasm expressed by some interviewees for PCNs as a sign of greater investment in the NHS in primary care, and as a way to raise the collective voice of GPs and primary care, for example, in terms of negotiating collective contracts. Some of those involved in the leadership and management of PCNs expressed that they have experienced a sense of empowerment in working on something larger than a single practice, and being involved in strategic planning of local primary and community health services over and above single-practice working.

*'The main thing that has come in—and this isn't just here— is the enthusiasm with which mostly a new set of GP faces have really taken on a new role and are invigorated and believe they're a bit empowered, and they're doing something at a bigger, more strategic level than out of practice'* (Int19)

Where PCNs are perceived as a continuation of existing efforts to improve GP sustainability and local healthcare, there seems to be a high level of enthusiasm and buy-in. However, where PCNs are perceived as a disruption to previous ways of working and a divergence from the goals of pre-existing forms of GP collaboration, there seem to be tensions and frustrations. On balance, engagement and buy-in will need to be fostered in order to build support for PCNs among wider primary care teams, and to ensure that those involved in managing and leading PCNs remain dedicated to their success.

The level of local engagement with and ownership of PCNs is connected to how they are structured as networks. Where PCNs are felt locally to be part of existing efforts to improve care, population health management and practice management, more individuals within GP practices appear to have bought into the premise of network working. In turn, the network is perceived to be founded on shared goals and interests, and less on top-down mechanisms that contractually bind network members together. However, the opposite is also true. Where PCNs are thought to be another top-down policy change, fewer individuals buy into the idea of PCNs, and there is likely to be increasing frustration about imposed interruptions to existing ways of working at a local level.

### The role of PCNs in the local health system

This evaluation explored the first year of the development and implementation of PCNs, as they were still finding their place within the wider health and social care system. Different local contexts, for example, relationships with statutory NHS bodies and histories of previous collaborative working, contributed to a diversity of ways in which PCNs have been working within local healthcare systems. As the COVID-19 pandemic emerged in 2020, this also influenced the role of PCNs within the local and wider NHS.

One way in which this variation played out was through the relationship between PCNs and local Clinical Commissioning Groups (CCGs), which commission most hospital and community services in local areas in England. Some PCNs had drawn on management support from the local CCG throughout their development and implementation, while other PCNs reported little involvement from the CCG, or even cases of tension where the CCG was perceived as exerting undue influence over PCN priorities and budgets.

Variation in local context was also evident in the relationship between PCNs and pre-existing forms of GP collaboration, including GP federations and super-partnerships. At times, PCNs had been able to build on good working relationships established from previous collaborative working between practices and with other parts of the health and social care system and voluntary sector, which helped establish the position of PCNs locally. In one case study site, the super-partnership exerted considerable influence on PCNs to which member practices belonged, to the extent that PCNs merged and expanded to fit the geographical boundary of the super-partnership. These shifts will inevitably affect an individual PCN's place within the local health and care system and the scale at which the PCN operates in terms of its patient population.

Lastly, the COVID-19 pandemic has further shaped the place of PCNs within local and national health and social care systems.[29] PCNs have been an important mechanism in delivering the national COVID-19 vaccination programme and have led the designation and deployment of vaccination sites after being asked to do so by NHS England and Improvement in December 2020.[30] Locally,

PCNs were key to organising the delivery of primary care during the pandemic, for example by organising 'hot' and 'cold' hubs to care for COVID-19 and non-COVID-19 patients, and helping to coordinate the movement of staff between practices.[13] PCNs' role in both national and local healthcare delivery during the pandemic has already influenced their role within the health and social care system (eg, by influencing national priorities that PCNs will focus on, including long COVID-19 and weight management),[31] in ways that will likely become clearer as England emerges from the pandemic.

The place of PCNs within the wider health and care system is also linked to how they are structured and gain legitimacy as networks. Depending on the PCN's relationship with other organisations locally and nationally (eg, with CCGs and local super-partnerships), and the demands being placed on PCNs due to system-level pressures (eg, the pandemic), the place of the PCN within the wider system shifts. At times, the PCN is a mechanism for collaboration on certain specified tasks, while at other times, it is a primary unit to deliver critical activities such as primary care's pandemic response, and a focal point for interaction between local primary care and other health and social care organisations.

## DISCUSSION

This evaluation reveals that PCNs, while introduced through national policy, are also based on shared goals of improving sustainability in primary care and improving integrated services for patients. While they are organised around delivering a set of priorities set out in the national DES contract,[28] they are also firmly based in local health and care systems, dependent on their local context and population health needs. Beyond their initial development and implementation, a challenge for PCNs

will be to build buy-in and engagement and clarify their place within the wider health and care system. To support PCNs as they continue to develop, and to ensure they are able to address both national priorities and local health population needs, including health inequalities, it will be important to ensure that appropriate management structures are in place, while also giving PCNs sufficient autonomy to adapt.

Although PCNs specifically are unique to the English NHS, thinking about what support they are likely to need to address local and national priorities longer term is informative for wider discussion of the international experience of meso-level primary care organisations. Primary care organisations in other jurisdictions find themselves, like PCNs, shifting between a focus on local and national health priorities, and face challenges finding their place in wider health and care systems. They also report the common risk of being swept into increasingly centralised functions such as those identified in national policy initiatives.[15]

Goodwin et al[21] provide a lens for thinking about the kind of management and support that PCNs and similar international examples of primary care organisation may need to ensure that they can reach their full potential. The authors established a typology of three types of networks, based on the level of social regulation and social integration within the network (see table 4).

PCNs can be understood both as enclave and hierarchical networks. They are simultaneously founded on shared goals and motivations and a relatively flat structure whereby each practice within the PCN has a voice, as well as being organised to be able to execute predefined tasks specified in the DES contract based on the national policy and funding infrastructure that initiated and surrounds them. Examining PCNs through this theoretical lens allows a more nuanced approach to the support that

**Table 4** Different networks structures—adapted from NHS Service Delivery and Organisation (SDO),[39] based on Goodwin et al[21]

| Network type | Key characteristics | Key lessons for network management |
|---|---|---|
| Enclave | ► High social regulation and low social integration. <br> ► Equality between members, flat internal structure. <br> ► High level of social cohesion and shared commitment to common interests, values and goals. | ► Creates bottom-up legitimacy and promotes creation of new ways of working. <br> ► May fail when motivation of members is exhausted or when tensions occur. <br> ► Management may be administrative, helping to facilitate collaborative working, but without formal audits. |
| Hierarchical | ► High social regulation and high social integration. <br> ► Centred on organisational core that is able to regulate its members. <br> ► May be sustained by common interests, values and goals, but also based on structured agreements and protocols. | ► Most successful in coordinating and executing predefined tasks. <br> ► May fail through over-regulation, which limits ability to innovate and leads to low motivation of members. <br> ► Management to coordinate defined activities and provide central direction, although it is suggested that mandated networks should be avoided. |
| Individualistic | ► Low social regulation and low social integration. <br> ► Single entities or organisations that come together to achieve certain tasks. <br> ► No strong sense of shared interests, values and goals. | ► Innovative and flexible, with fluid membership. <br> ► May fail due to high cost of membership, competition and conflict between members that can limit desire to work jointly. <br> ► Management may help set targets, incentives and monitoring activities. |

NHS, National Health Service; SDO, Service Delivery and Organisation.

PCNs will require going forward, including in addressing the issues that PCNs face in terms of securing local ownership and engagement, and clarifying their role within the wider health and social care system.

As enclave networks, PCNs share the common goal of wanting to ensure sustainability in primary care, including financial and workforce sustainability, and improving integrated services that meet the needs of the patients of constituent practices. Locally, there is a preference for focusing on the characteristics that PCNs share with enclave networks, as evidenced by the enthusiasm and commitment that was expressed for the underlying goals of PCNs and the ability to work collaboratively to address local population health needs. This contrasted with the reticence and frustration towards the top-down, prescriptive nature of PCN policy, particularly where PCNs were not perceived to be aligned with local priorities. Fostering this sense of shared goals and intrinsic motivation may help encourage buy-in and engagement with PCNs, and allow them the space and autonomy to arrive at solutions that address local population health needs. Even as PCNs continue to address national health priorities and complete pre-defined tasks, it will be important to balance and align these with local priorities to foster buy-in, engagement and a shared sense of interests and goals within PCNs.

PCNs also share characteristics with hierarchical networks—they emerged from a centrally determined policy and funding mechanism, and are designed to deliver services as set out in the national specification for PCNs.[28] In this sense, PCNs are well suited to deliver on predetermined tasks and respond to direction and guidance from central bodies, and have been effective in quickly making progress towards national strategic goals by establishing new and enhanced services for patients. However, as hierarchical networks they face a risk of over-regulation and excessive performance management that could inhibit motivation and enthusiasm for PCN teams and hamper their ability to innovate locally, which has been an issue for predecessor primary care organisations.[32 33]

The risk of over-regulation will be especially important to consider as the proposed Integrated Care Systems (ICSs) are implemented nationally, CCGs are abolished, and PCNs likely find themselves having to work out their role within a restructured NHS.[34] PCNs have been identified as critical to the future success of ICSs by NHS England and Improvement and the Department of Health and Social Care,[27] which will likely have implications in terms of how PCNs are organised. It is possible that PCNs will come under pressure to grow in size and complexity, merge with neighbouring PCNs, which will add to the challenges they face in terms of local engagement if these risks are not carefully mitigated. The risk that PCNs are increasingly drawn into formal hierarchical arrangements and mergers is a common experience among meso-level organisations in primary care in the international context.[35–37]

## CONCLUSION

This evaluation reveals that PCNs demonstrate significant potential to swiftly deliver new services to patients, respond to national priorities, bring together primary care providers with common motivations and interests, and improve financial and workforce sustainability in primary care. Furthermore, during the pandemic PCNs have responded to both national priorities in their participation in England's vaccination programme, as well as responding rapidly to local needs, for example, by coordinating the movement of staff and patients between 'hot' and 'cold' hubs.

The task ahead for PCNs will be to ensure that they are able to address national priorities that are centrally defined, as well as adapting to fit local health needs. Focusing on the shared goals that motivate GP practices to want to collaborate with one another, and protecting PCNs from over-regulation, will be especially important as PCNs find their place within the wider NHS as it emerges from the pandemic, and as ICSs are implemented.

Primary care organisations like PCNs are often strongly placed to address local and national needs, being both enclave and hierarchical in nature, and should continue to address both of these areas. Careful attention needs to be paid to how these priorities are balanced, and how decisions are made that shape how these organisations fit into wider health and care systems. In order to enable these organisations to reach their full potential, the core characteristics of these organisations must be considered in deciding how they should be managed, including the motivations driving individual providers to join these organisations and the policy context that led to their development.

**Acknowledgements** We thank Amelia Harshfield, who contributed to the conception and design of the study, and data collection at two study sites. We also thank Natasha Elmore, Dr Sarah Ball, and Jon Sussex (all from RAND Europe) for their contribution to data collection, gaining ethical approval, and providing project advice; Dr Rebecca Fisher (Senior Policy Fellow, Health Foundation) and Dr Mina Gupta (GP, Modality Partnership) for peer review of our online surveys; Samantha Hinks (NHS England and Improvement) and Professor Katherine Checkland (University of Manchester) for their on-going advice and sharing learning from their own evaluations and research throughout; Professor Russell Mannion (University of Birmingham), Dr Katie Coleman and Mark Platt (BRACE Centre Health and Care Panel) who reviewed our study protocol; and Professor Justin Waring (University of Birmingham) and Dr Anna Dixon (formerly Chief Executive, Centre for Ageing Better) for undertaking critical review of our findings.

**Contributors** JS was principal investigator for the study and responsible for its conception, design, conduct and writing up, with support from MS and SP. MS, SP and JS undertook data collection and SP led data analysis supported by JS and MS. SP wrote the first draft of this paper, and MS and JS commented on and contributed to drafts. JS is the guarantor for this work, and is responsible for the overall content.

**Funding** This project was funded by part of a grant from The National Institute for Health Research, Health Services and Delivery Research programme (HSDR 16/138/31 – Birmingham, RAND and Cambridge Evaluation Centre).

**Competing interests** None declared.

**Patient and public involvement** Patients and/or the public were involved in the design, or conduct, or reporting, or dissemination plans of this research. Refer to the Methods section for further details.

**Patient consent for publication** Not applicable.

**Ethics approval**  This study involves human participants and received ethical approval from the University of Birmingham Research Ethics Committee (ERN-13-1085AP34).

**Provenance and peer review**  Not commissioned; externally peer reviewed.

**Data availability statement**  No data are available. Due to the consent process for data collection at case study sites within this evaluation, there are no data that can be shared.

**ORCID iDs**
Sarah Parkinson http://orcid.org/0000-0002-2858-1842
Judith Smith http://orcid.org/0000-0003-4036-4063
Manbinder Sidhu http://orcid.org/0000-0001-5663-107X

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
