## [Reviewer comments · BMJ Open]

ARTICLE DETAILS

TITLE (PROVISIONAL)	The early development of primary care networks in the NHS in England: A qualitative mixed-methods evaluation
AUTHORS	Parkinson, Sarah; Smith, Judith; Sidhu, Manbinder

VERSION 1 – REVIEW

REVIEWER	Stokes, Tim Otago University, General Practice & Rural Health
REVIEW RETURNED	06-Sep-2021

GENERAL COMMENTS	This paper reports a mixed methods rapid evaluation of the early development of primary care networks (PCNs) in the English NHS. PCNs are an important aspect of recent NHS reforms, and an evaluation of their early development is both important and timely. Overall this is an excellent paper. As highlighted in the strengths and limitations box not only are the evaluation methods highly appropriate, but the authors are to be commended for using relevant organisational theory (health networks) to better understand and draw insights from the collected data. I have no suggested major or minor revisions to the paper. Minor editorial comments 1. p.8, LL45. Fix "Error Ref not found"2. p.14, Table 4. "Adapted" rather than "Adopted"?3. The references need reviewing with relevant hyperlinks, e.g., refs 31 and 324. Insertion of the Squire 2 reporting standards. I am unclear what this adds to the paper - does it need including? This may be more of question to the BMJ Open team.
---

REVIEWER	Rowe, Rachael NHS Somerset Clinical Commissioning Group, Clinical Commissioning
REVIEW RETURNED	10-Sep-2021

GENERAL COMMENTS	Thanks you for this interesting and very timely paper. The sample size is mainly rural. I can see that this is limited by the application and selection method but it raises questions whether there are differences in how more urban inner city practices are implementing PCNs and their challenges. A well structured paper raising some very pertinent issues around
---

VERSION 1 – AUTHOR RESPONSE

Reviewer: 1

Prof. Tim Stokes, Otago University

Comments to the Author:

This paper reports a mixed methods rapid evaluation of the early development of primary care networks (PCNs) in the English NHS. PCNs are an important aspect of recent NHS reforms, and an evaluation of their early development is both important and timely.

Overall this is an excellent paper. As highlighted in the strengths and limitations box not only are the evaluation methods highly appropriate, but the authors are to be commended for using relevant organisational theory (health networks) to better understand and draw insights from the collected data.

I have no suggested major or minor revisions to the paper.

RESPONSE: Thank you very much for your feedback, and we are thrilled about the opportunity to publish this timely paper. We feel that organisational theory around health networks (adapted from Goodwin and colleagues) really helped us understand what sort of networks primary care networks are, and how this related to what type of support they will need going forward.

Minor editorial comments

1. p.8, LL45. Fix "Error Ref not found"

RESPONSE: Thank you -- this has now been corrected.

2. p.14, Table 4. "Adapted" rather than "Adopted"?

RESPONSE: Thank you -- this has now been corrected and should have read adapted.

3. The references need reviewing with relevant hyperlinks, e.g., refs 31 and 32

RESPONSE: The reference list has been reviewed and edited to include up to date hyperlinks and DOIs where possible.

4. Insertion of the Squire 2 reporting standards. I am unclear what this adds to the paper - does it need including? This may be more of question to the BMJ Open team. [EDITOR: BMJ Open requires authors to complete the relevant reporting checklist]

RESPONSE: A completed Squire 2.0 checklist has now been uploaded along with the manuscript.

Reviewer: 2

Miss Rachael Rowe, NHS Somerset Clinical Commissioning Group

Comments to the Author:

Thanks you for this interesting and very timely paper.

The sample size is mainly rural. I can see that this is limited by the application and selection method but it raises questions whether there are differences in how more urban inner city practices are implementing PCNs and their challenges.

A well structured paper raising some very pertinent issues around primary care networks and their future role.

RESPONSE: Thank you for your feedback -- we are pleased to hear that reviewers found the paper important and timely. We agree that there are important questions to still be considered regarding differences between rural and more urban primary care networks. We consider this in an NIHR report for this study by the same authors, which is cited in the report, and hope to see more engagement with this topic in the literature in the future.